# School-Based Team Sports as Catalysts for Holistic Student Wellness: A Narrative Review

**DOI:** 10.3390/bs14070528

**Published:** 2024-06-25

**Authors:** Xiaolei Kang, Qing Meng, Chun-Hsien Su

**Affiliations:** 1School of Physical Education and Humanities, Nanjing Sport Institute, Nanjing 210014, China; 2010070012@nsi.edu.cn; 2School of Physical Education, Huaqiao University, Xiamen 361021, China; mq@hqu.edu.cn; 3Sport and Health Research Center, Huaqiao University, Xiamen 361021, China; 4Department of Exercise and Health Promotion, Chinese Culture University, Taipei City 111369, Taiwan; 5College of Kinesiology and Health, Chinese Culture University, Taipei City 111369, Taiwan

**Keywords:** holistic development, physical activity benefits, educational strategies, wellness frameworks, student engagement, inclusive sports programs

## Abstract

The narrative review examines how school-based team sports catalyze holistic student wellness, leveraging their inherent nature and addressing barriers to inclusivity. Utilizing the holistic wellness framework—which encompasses physical, emotional, intellectual, social, spiritual, and occupational dimensions—the review evaluates the multifaceted effects of these sports on student well-being. Supported by wellness models like Dr. Bill Hettler’s Six Dimensions of Wellness and various research studies, the framework highlights the interconnectedness of these areas in achieving overall well-being. School-based team sports promote discipline, teamwork, physical fitness, and social interaction while fostering skill development, ethical behavior, and emotional resilience. These elements, collectively, may support the holistic development of students, enhancing their overall health and success. The methodological approach of this review involves a synthesis of empirical research, systematic reviews, and meta-analyses from the past two decades (2004–2024), sourced from databases such as PubMed, EBSCO (EDS), and Google Scholar. Key findings include enhanced cardiovascular fitness, emotional resilience, cognitive performance, social skills, spiritual fulfillment, and career readiness awareness. The review highlights the necessity of integrating team sports into educational curricula to promote well-rounded student development and proposes strategies to overcome socioeconomic, cultural, and structural barriers to participation. Future research should focus on longitudinal studies to examine the long-term effects of team sports and explore the potential of digital technologies and artificial intelligence (AI) in enhancing these benefits.

## 1. Introduction

### 1.1. Understanding Holistic Wellness: A Framework for Holistic Student Development

Recent academic perspectives have expanded the definition of wellness to encompass a holistic view, integrating physical, emotional, intellectual, social, spiritual, and occupational dimensions [1]. The foundation for incorporating these dimensions within the holistic wellness framework is supported by numerous wellness models and empirical studies that underscore the interconnectedness of these areas in achieving overall well-being. The Six Dimensions of Wellness by Dr. Bill Hettler of the National Wellness Institute is noteworthy among these models, initially including physical, emotional, intellectual, social, spiritual, and occupational dimensions [2]. The Global Wellness Institute articulates wellness as the active pursuit of activities, choices, and lifestyles conducive to holistic health, emphasizing self-responsibility across these dimensions. In contrast to healthcare, which primarily focuses on reactive disease treatment, wellness prioritizes proactive disease prevention and attaining optimal well-being [2].

Physical wellness refers to maintaining a healthy body through regular physical activity, balanced nutrition, sufficient rest, and avoiding harmful behaviors. It emphasizes the importance of exercise to enhance cardiovascular health, muscle strength, flexibility, and endurance while focusing on practices that prevent illness and promote overall bodily function and vitality. Emotional wellness involves understanding and managing one’s emotions, maintaining a positive outlook, and effectively coping with stress. It includes recognizing and accepting feelings, seeking needed support, and practicing self-care to promote emotional resilience and mental health. Intellectual wellness involves engaging in mentally stimulating activities, expanding knowledge, and fostering critical thinking and creativity. It includes pursuing lifelong learning, seeking out new experiences and challenges, and maintaining curiosity and an open mind to enhance cognitive function and personal growth. Social wellness emphasizes the importance of developing and maintaining healthy relationships, effective communication, and a supportive social network. It involves building strong connections with family, friends, and the community, engaging in social activities, and contributing to the well-being of others to foster a sense of belonging and social stability. Spiritual wellness involves seeking meaning and purpose in life through various means, such as religion, meditation, or a connection with nature. It includes exploring personal beliefs and values, experiencing inner peace, and aligning actions with those values to achieve a sense of harmony and fulfillment. Occupational wellness focuses on achieving personal satisfaction and enrichment through work or academic endeavors. It involves balancing work and leisure, pursuing career goals that align with personal values, and developing skills and strengths to achieve fulfillment and a sense of purpose in one’s professional life [1,2,3].

These dimensions collectively offer a comprehensive approach to achieving a balanced and fulfilling life. The holistic wellness framework embodies this broad perspective, categorizing wellness into essential dimensions critical for holistic health. In both academic and practical applications, this comprehensive approach ensures a well-rounded understanding of wellness, addressing the diverse aspects of students’ lives that contribute to their overall health and success. Holistic wellness is crucial for achieving balance and fulfillment for students grappling with academic pressures, personal development challenges, and career uncertainties [3]. By adopting a holistic wellness perspective, the multifaceted aspects of student success and well-being are acknowledged, extending beyond academic achievements to physical health, emotional resilience, and social connections, thereby promoting a holistic approach to student development [4].

### 1.2. Rationale for School-Based Team Sports

School-based team sports programs focus on joining students together on the same team. There are many options for playing school-based sports, including opportunities for pre-K to 8th grade, high school, and college participants. School-based team sports foster a structured environment that promotes discipline, teamwork, physical fitness, and social interaction while encouraging skill development, ethical behavior, and emotional resilience. These elements collectively contribute to the holistic development of students, supporting their overall health, success, and well-being. School-based team sports involve participants working together to achieve common goals and are instrumental in developing communication, conflict management, and problem-solving skills within a trusting team environment. Research indicates that basketball, soccer, volleyball, and handball benefit elementary, middle school, and collegiate students by enhancing social skills, academic performance, and leadership abilities. These sports promote physical and psychological well-being, which are essential for future teamwork in professional settings. This article analyzes and discusses these aspects, supported by a growing body of research that positions school-based team sports as a catalyst for enhancing student wellness across the holistic wellness dimensions [5,6].

School-based team sports facilitate physical exercise and provide dynamic platforms for emotional support, intellectual engagement, and social integration. For instance, recent research [7,8] has found that participation in team sports is associated with improved physical health outcomes, such as increased cardiovascular fitness and reduced obesity rates, highlighting the direct benefits of physical activity on health. Moreover, school-based team sports enhance emotional well-being by fostering community and belonging, effectively reducing stress and anxiety [9]. The cognitive demands of school-based team sports, including strategic planning and quick decision-making, are linked to improved academic performance and cognitive function, underscoring the intellectual benefits of these activities [10]. Socially, school-based team sports develop essential life skills, such as communication and leadership, contributing to more robust social networks and enhanced interpersonal skills [11]. Participating in school-based team sports also contributes to spiritual wellness by giving individuals a deep purpose and fostering connections with a broader community. This engagement enhances spiritual health through a strengthened sense of belonging, unity, and commitment to common goals, enriching participants’ spiritual lives [12]. On the occupational front, the discipline and teamwork developed through school-based team sports translate into valuable skills that enhance collaboration and work ethic. These skills are crucial in academic and professional settings, providing participants with a competitive advantage and a flexible skill set highly valued in today’s workforce [13].

Integrating school-based team sports into wellness programs and educational curricula enables schools and universities to harness these multidimensional benefits, supporting student development in alignment with holistic wellness dimensions. This approach facilitates well-being across all aspects of students’ lives and cultivates academic proficiency, physical health, emotional resilience, social adeptness, spiritual fulfillment, and occupational preparedness [7,8,9,10,11,12,13,14]. This review aims to elucidate the comprehensive impacts of school-based team sports on students’ holistic wellness, as defined by the holistic wellness dimensions. Recognizing the benefits of and identifying the barriers to participation, we seek to enhance inclusivity and accessibility in school-based team sports within educational settings.

## 2. Methodology

### 2.1. Research Design and Criteria for Literature Selection

This comprehensive review rigorously examines the multifaceted effects of school-based team sports on student wellness using the holistic wellness framework, which encompasses physical, emotional, intellectual, social, spiritual, and occupational dimensions of well-being. We employed a narrative synthesis methodology to integrate and analyze findings from diverse studies. Our systematic approach to literature selection focused on empirical research, including quantitative and qualitative studies, to thoroughly investigate the impacts of school-based team sports on various wellness dimensions. Additionally, we reviewed existing reviews, systematic reviews, and meta-analyses to provide a holistic examination of the subject matter. The findings of this review are expected to significantly enhance understanding of how school-based team sports affect student wellness, contributing valuable insights to the fields of sports science, public health, and wellness.

### 2.2. Search Strategy Description

A comprehensive literature search was conducted across several academic databases, including PubMed, EBSCO (EDS), and Google Scholar. We used the following search terms: ‘(team sports) AND (wellness OR well-being) AND (physical wellness OR emotional wellness OR intellectual wellness OR social wellness OR spiritual wellness OR occupational wellness)’. The findings from the selected literature were synthesized using a narrative approach, integrating insights from various studies to draw broad conclusions about the effects of school-based team sports on student wellness. The detailed search strategy is presented in Figure 1 to enhance transparency and credibility.

### 2.3. Roles of Authors and Conflict Resolution

Each author was assigned specific roles: initiating the search, screening the articles, extracting data, and drafting the manuscript sections. Disagreements among authors regarding study inclusion were resolved through discussion until consensus was achieved. If consensus could not be reached, a third-party expert in exercise physiology was consulted to make the final decision.

### 2.4. Selection and Inclusion Criteria

The initial selection process began with a pool of 829 studies, which underwent a rigorous selection phase based on specific inclusion criteria applied through three successive filters to refine the final sample:

First Filter—Before-Screening Level:

Objective: Remove duplicate records.

Outcome: Reduced the initial pool by eliminating 387 duplicate records, leaving 442 studies.

Second Filter—Abstract Level:

Criteria:

Studies employing quantitative and qualitative research methods.

Involvement of team sport or wellness.

Open-access or peer-reviewed.

Published between 2004 and 2024.

Written in English.

Outcome: Screened 442 abstracts, excluding 319 studies that did not meet the criteria. A total of 123 studies moved to the next stage.

Third Filter—Full-Text Level:

Criteria:

Studies featuring systematic reviews, meta-analyses, detailed statistical measurements, or review studies relevant to the subject matter.

Exclusion of editorials, brief reports, communications, perspectives, concept papers, and opinions.

Target population involving children, adolescents, or collegiate students.

Outcome: Reviewed 123 full-text articles, excluding 83 that did not meet the criteria. A final sample of 40 studies was selected for this comprehensive review. 

The thorough and structured selection process ensured the inclusion of high-quality studies that provided robust evidence for the review’s conclusions. This detailed methodology enhances the transparency and reproducibility of our review.

## 3. The Multidimensional Impact of School-Based Team Sports: Exploring the Benefits of School-Based Team Sports across Wellness Domains

School-based team sports stand out as a potent catalyst for enhancing wellness across multiple domains of the wellness wheel. By engaging students in structured physical activity within a team setting, these sports offer benefits beyond the physical to encompass emotional, intellectual, social, spiritual, and occupational wellness. This section delves into the empirical evidence and theoretical foundations underlying the multidimensional impact of school-based team sports on student wellness.

### 3.1. Physical Wellness: Strengthening the Body through School-Based Team Sports

School-based team sports significantly enhance cardiovascular fitness, muscular strength, and body composition through aerobic and anaerobic activities [15]. For instance, studies highlight the benefits of specific anaerobic training routines in improving heart rate and exercise intensity, emphasizing the need for optimized training regimens [16,17].

Participation in these sports improves health indicators such as blood pressure, lipid profiles, and glucose control, reducing risks of chronic diseases like heart disease, diabetes, and obesity. Outcomes include reduced systolic blood pressure and interleukin-6 levels and improved body composition with decreased fat mass and increased lean muscle mass [15,18,19]. These findings underscore the role of team sports in chronic disease prevention and management. Compared to solitary sports, school-based team sports enhance health and motivate sustained physical activity [20]. The social interaction in team sports is crucial for maintaining exercise routines. Incorporating these sports into educational programs shows improvements in body composition, running ability, muscle strength, heart function, and flexibility [21,22], supporting their inclusion in school curricula [23,24].

The enjoyment and variety of school-based team sports encourage long-term engagement. The pleasure from participation and diverse challenges increases ongoing physical activity compared to solitary exercise, highlighting the importance of promoting these sports for lifelong physical wellness [25]. School-based team sports inherently promote social interaction and enjoyment, leading to sustained participation in physical activity. This continuous engagement enhances health and reduces the risk of chronic diseases. Integrating these sports into educational programs is essential for comprehensive student health and wellness. Table 1 summarizes the principal outcomes regarding the influence of school-based team sports on physical wellness, with references to corroborative studies.

### 3.2. Emotional Wellness: Cultivating Resilience through Team Participation

Engaging in school-based team sports provides significant mental health benefits, including reduced depression, anxiety, and stress. These benefits stem from the social support systems within these sports, which enhance feelings of belonging and emotional well-being. A systematic review found that team sports improve self-esteem and social interaction and reduce depressive symptoms more effectively than individual activities [26,27].

A mixed-method study showed that participation in team sports correlates with lower depressive symptoms and higher life satisfaction for both genders. Boys preferred sports like football and cricket, while girls favored activities such as skipping and running. The findings emphasized that school-based team sports and athlete identity are linked to lower depression scores, highlighting the role of social cohesion in emotional wellness [26,28,29]. Team sports also improve self-control, foster pro-social behaviors, and nurture a sense of community [30]. Team dynamics and challenges help develop emotional competencies like empathy, emotional regulation, and resilience, supporting comprehensive emotional growth.

Another study demonstrated that sports participation predicts lower social anxiety and loneliness, especially in adolescents with higher pre-existing psychological difficulties. Both team and individual sports are associated with reduced depression in boys, while team sports specifically reduce depression in girls [31,32]. This suggests that social interaction in team sports particularly benefits girls’ mental health. Gender-specific preferences and obstacles in sports engagement, such as screen time, academic demands, cultural expectations, and lack of facilities, must be addressed to maximize the benefits [33]. Additionally, school-based team sports and sports clubs provide vital social and psychological support for mental health [34,35]. Research on youth athletes revealed that strong group identification enhances self-worth, commitment, and effort [36,37]. Boys in team sports exhibited fewer mental health issues than inactive peers, underscoring the positive impact of team sports on mental health [32].

School-based team sports provide essential social and psychological support, improving cognitive function and emotion regulation. These benefits highlight the importance of incorporating team sports into school curricula to support students’ holistic development. Table 2 summarizes the main findings on the effects of school-based team sports on emotional wellness, supported by references.

### 3.3. Intellectual Wellness: Sharpening the Mind through School-Based Team Sports

School-based team sports extend their benefits into the intellectual domain, enhancing cognitive functions and contributing to academic achievement. These sports act as both physical and mental exercises, fostering various intellectual skills.

Research demonstrates a positive impact of sports participation on academic performance. A study found that sports activities improve academic outcomes, mediated by gains in cardiorespiratory fitness and complex motor skills directly linked to academic improvement [38]. Additionally, exercise and group activities positively affect cognitive performance, especially for boys, suggesting that while exercise is crucial for boys, other factors in group membership might be more influential for girls [39].

Data indicated that physical activity and sports team participation were associated with higher GPAs for high school girls, while only sports team participation was linked to higher GPAs for high school boys. For middle school students, the positive association between physical activity and GPA was intertwined with sports team participation [40]. A systematic literature review on young athletes found a positive relationship between sports participation and academic performance. Despite the demands of a dual career, young athletes perceive more benefits than harms from sports, contributing to their overall growth. Further research is needed to understand the long-term relationship between sports training and student-athletes’ academic and personal lives [41].

These findings support the inclusion of school-based team sports in educational programs to enhance academic outcomes. The evidence suggests that physical and cognitive engagement through sports improves academic performance and develops critical cognitive skills. Further research should explore the specific mechanisms and long-term impacts on academic and career success [38,39,40,41]. Table 3 summarizes the main findings on the contribution of school-based team sports to intellectual well-being, including enhancements in cognitive abilities and academic success, supported by references.

### 3.4. Social Wellness: Building Community through School-Based Team Sports

School-based team sports significantly enhance social wellness by providing a structured environment for developing social skills, building relationships, and fostering community.

A systematic review of sports programs for socially vulnerable youth found that sports participation enhances cognitive and social life skills, though few studies explored the specific conditions for optimal life skill development [42]. Another study on a team-sports-based life-skills program during physical education showed improvements in sport skills tests and life skills knowledge, indicating the effectiveness of structured programs [43]. Research on the social environment of sports teams revealed that enjoyment mediated the relationship between social identity and sports dropout among adolescent girls, emphasizing the importance of fostering a positive team environment to retain participants [44]. Additionally, youth sports provide social benefits and address broader social issues, highlighting the need for intentional program design to maximize these benefits [45]. Adolescents active in sports show greater school involvement and peer acceptance and are less likely to engage in risky behaviors, underscoring the role of team sports in fostering a supportive social milieu [30,42,44,45]. School-based team sports also promote social cohesion and community integration, bridging cultural gaps and enhancing social assimilation [46,47].

These findings support the inclusion of school-based team sports in educational curricula to maximize their social benefits. Table 4 summarizes the principal insights regarding the effects of school-based team sports on social wellness, supported by references.

### 3.5. Spiritual Wellness: Finding Purpose in School-Based Team Sports

School-based team sports significantly enhance spiritual wellness by fostering shared goals and values, creating a sense of purpose and connection.

A study examined the relationship between team sports participation, depression, and suicidal ideation among adolescents, focusing on differences between heterosexual, LGBQ, cisgender, and transgender youth. It was found that sports participation reduced depression across all groups and suicidal ideation in all groups except LGBQ youth, highlighting the need for inclusive sports policies to support all youth [48]. Participation in sports is linked to personal and spiritual growth, offering pathways to self-transcendence and excellence. Research revealed higher levels of spirituality among athletes compared to the general student population, with variations by gender and ethnicity. Female and White athletes reported higher spirituality scores than their male and Black counterparts, emphasizing the role of higher education institutions in fostering spiritual development [49]. Spirituality enhances sporting performance, personal growth, and well-being. Despite its importance, spiritual well-being has received limited attention in sports and exercise psychology. It is advocated that spiritual well-being be incorporated into sports psychology training, identifying areas for future research [50]. A study on college student-athletes perceptions of spiritual care in athletic training revealed that respondents believe spirituality can influence treatment progress and lead to positive outcomes. They suggested that athletic trainers should have basic skills to support athletes’ spiritual needs [51].

These findings support the inclusion of spiritual wellness considerations in school-based team sports programs to enhance overall well-being. Incorporating spiritual wellness in these programs can help students develop a stronger sense of purpose and meaning in their lives. Schools can help students achieve a more balanced and fulfilling educational experience by fostering an environment that supports spiritual growth. Table 5 summarizes the critical findings on the impact of school-based team sports on spiritual wellness, supported by references.

### 3.6. Occupational Wellness: Preparing for the Future with School-Based Team Sports

School-based team sports foster essential life skills such as teamwork, leadership, and problem-solving, which are crucial for academic and career success. These sports develop personal and professional competencies, preparing individuals for future challenges.

Recent literature highlights the role of sports in developing entrepreneurship, attracting youth to entrepreneurial education, and providing specialized support to sports entrepreneurs. These strategies underline the potential of sports in entrepreneurship education [52,53]. A study found that male students had higher goal-setting skills, and those with more sports experience and team participation showed superior life skills. Participants in school-based team sports scored higher in time management, leadership, teamwork, and goal setting than in individual sports [54].

These findings support the inclusion of school-based team sports in educational curricula to enhance occupational wellness. Team sports prepare students for professional success and personal growth by fostering critical life skills and promoting entrepreneurship. Table 6 summarizes the primary outcomes related to how participation in school-based team sports affects different elements of occupational wellness, supported by literature references.

### 3.7. Interconnected Wellness: Exploring the Multidimensional Impact of School-Based Team Sports on Student Development

School-based team sports significantly influence various dimensions of student development, encompassing physical, emotional, intellectual, social, spiritual, and occupational wellness [15,18,27,38,43,49,52]. The collaborative nature of these sports fosters essential life skills such as teamwork, leadership, and problem-solving, preparing individuals for academic success and future career challenges [40,53]. Participation in sports also promotes mental health and emotional well-being, providing a supportive social environment that enhances students’ sense of belonging and community [33,35]. Furthermore, school-based team sports contribute to spiritual wellness by encouraging responsible behaviors and fostering a deeper connection with nature [48].

This holistic approach emphasizes integrating school-based team sports into educational programs to cultivate well-rounded, resilient, and socially responsible individuals. These programs enhance physical fitness by promoting a comprehensive development framework and instilling values and skills crucial for personal and professional success. Including spiritual aspects further enriches students’ educational experiences, guiding them toward becoming more conscientious and engaged citizens. Integrating school-based team sports into educational curricula is essential for fostering students’ overall well-being and development. This approach supports the formation of a balanced, resilient, and socially responsible individual equipped to meet future challenges effectively.

## 4. Overcoming Barriers: Enhancing Access and Inclusivity in School-Based Team Sports

The potential of school-based team sports to enhance students’ multidimensional wellness is clear. However, realizing this potential requires addressing barriers such as socioeconomic constraints, cultural hurdles, and structural issues that restrict access and inclusivity. The sub-categories in this section were derived from a thematic literature analysis, identifying recurring themes related to enhancing access and inclusivity. These themes were categorized into policy and infrastructure, socioeconomic disparities, gender equality, disability support, cultural competence, and community engagement.

### 4.1. Addressing Socioeconomic Disparities: Ensuring Equal Opportunities for All Students

Socioeconomic disparities significantly impact youth participation in physical activity and sports, with children from low-affluence families engaging less frequently and participating in fewer sports than their more affluent peers. Barriers include fear of injury, unwelcoming team environments, high costs, and transportation issues [55]. Middle schoolers from affluent families are three times more likely to meet physical activity recommendations, and high schoolers are similarly likelier to participate in sports than their low-affluent counterparts [55].

To foster inclusivity, sports organizations must address these barriers. For example, improvements to financial assistance programs based on parent feedback highlight the need to recognize and remove unintentional barriers [56]. Addressing economic and logistical obstacles can enhance participation across socioeconomic lines.

Healthcare disparities in rural and minority populations can be mitigated through initiatives that provide free sports/school physicals to middle- and high-school students, address health issues, and reduce financial burdens [57]. Such initiatives demonstrate the importance of providing necessary health services to promote equitable access to sports programs.

Creating inclusive environments requires maintaining well-equipped facilities, raising awareness about the benefits of sports, and providing transportation solutions. Educational institutions and community organizations must proactively engage excluded students and their families. By addressing these barriers, school-based team sports can become more equitable, contributing to all students’ holistic development and overall well-being.

### 4.2. Promoting Gender Equality: Breaking down Stereotypes and Encouraging Participation

Promoting gender equality in school-based team sports fosters inclusivity, breaks traditional gender stereotypes, and enhances health, fitness, and social interaction while upholding human rights. However, sports often fail to deliver these benefits, particularly for women and transgender athletes who face gender-based discrimination. This section examines instances of such inequality and evaluates two Canadian policy responses, highlighting the inadequacies of existing policies. It emphasizes the need for a stronger focus on sport’s foundational values and mechanisms to create an inclusive culture [58].

Research on gender and sexual diversity in sports underscores the necessity of inclusive sports cultures. A scoping review identified vital themes: identity, discrimination, coming out, the body, and strategies for social change. The review calls for more research on bisexual, transgender, and intersex athletes, GSM coaches, junior athletes, and the intersection of gender and sexuality with other identities. Combining anti-discrimination policies with gender and sexual-diversity education could positively impact GSM athletes and coaches [59].

Current policies often lack robust enforcement mechanisms. Sports organizations must adopt comprehensive anti-discrimination policies supported by education and training on gender and sexual diversity. Showcasing diverse role models and monitoring participation rates are essential for promoting gender equality and creating an inclusive environment in educational and sports organizations. These measures ensure that school-based team sports are equitable, benefiting all participants and fostering holistic wellness.

### 4.3. Supporting Students with Disabilities: Adapting Sports Programs for Inclusive Participation

Inclusive sports programs are essential for providing equal opportunities for students with disabilities to engage in school-based team sports. Research indicates that adapted sports significantly enhance well-being, resilience, social support, personal development, quality of life, and societal integration for individuals with disabilities, highlighting their importance [60].

A study on a modified tennis program, which used low-compression balls and promoted tennis as “easy, fun, and healthy”, showed increased participation and positive attitudes among coaches and national associations. These strategies can be effectively applied to other sports to improve participation and skill development [61].

Another study focused on community–academic partnerships in adaptive sports and recreation (ASR) to address local opportunities for individuals with disabilities (IDs). It identified a lack of knowledge about ASR as a barrier. The partnerships led to a community event to raise awareness and connect organizations. Survey results highlighted the need for targeted engagement strategies to address the specific needs of different groups [62].

To create a supportive environment, training coaches and staff in adaptive sports techniques and disability awareness is crucial. Peer mentoring fosters social integration and mutual respect among students. Securing funding through grants, donations, and partnerships is vital for sustaining and expanding these programs, promoting physical health, social integration, and emotional well-being for all participants.

### 4.4. Cultural Competence in Sports: Fostering Diversity and Inclusion

Enhancing cultural competence in sports is crucial for fostering diversity and inclusion, ensuring all students feel welcomed and valued. Strategies to promote intercultural competence include fostering intercultural friendships, organizing study abroad programs, facilitating Internet-based intercultural contact, establishing school-community partnerships, and encouraging critical reflection on intercultural experiences. Additional methods include cooperative and project-based learning, role plays, simulations, text and film analysis, ethnographic tasks, a culturally inclusive curriculum, and a whole-school approach to diversity and human rights [63].

In British sports culture, meritocracy and fairness have driven social equality policies since the late 1990s, with entities like UK Sport, the English Football Association (FA), and Kick It Out promoting inclusivity and diversity. Continuous evaluation is needed to meet diverse community needs [64]. Research highlights the under-representation of culturally diverse women in sports leadership, identifying themes like personal characteristics, interpersonal support, organizational support, and discrimination affecting Black or African American women in US college athletics, indicating a need for further research to enhance diversity and inclusivity [65].

Creating an inclusive environment requires resources and support for students from diverse backgrounds, including language assistance and culturally relevant coaching methods. Continuous assessment and adaptation of these strategies are essential for meaningful and lasting change in promoting diversity and inclusion in sports.

### 4.5. Community Engagement: Building Support Networks to Enhance Accessibility

Community engagement is essential for enhancing accessibility to school-based team sports and fostering inclusivity. Research underscores the role of sports in building community on college campuses, typically managed by student-affairs personnel. Despite this, detailed insights into the mechanisms remain limited. A qualitative approach identified common interests, leadership opportunities, voluntary activity, and competition as key factors in nurturing community within sports clubs, offering practical implications for sports managers and administrators [66].

A university–school district partnership increased fourth-grade students’ awareness of college and enhanced university student-athletes’ understanding of community needs, improving student motivation and reducing behavioral issues [67]. Additionally, a systematic review highlighted the role of sports volunteers in boosting community participation by motivating and facilitating engagement, underscoring the importance of volunteer efforts in expanding sports programs’ impact [68].

To enhance accessibility, developing transportation solutions and maintaining accessible facilities are vital. Promoting cultural and social events centered around sports further strengthens community bonds. This collaborative approach fosters inclusivity and holistic student development. Continuous evaluation and adaptation of these strategies ensures they meet the diverse needs of the student population, maximizing the benefits of school-based team sports for community building and student well-being.

### 4.6. Policy and Infrastructure: Creating Inclusive Environments in Educational Institutions

Establishing inclusive policies and infrastructure is crucial for equitable access to school-based team sports, especially for students with disabilities. Policies must address discrimination and unequal treatment based on disability and cultural background, with clear guidelines for reporting [69].

Participatory action research demonstrates the benefits of interventions like reconfiguring learning content, peer teaching, and discussions on inequity and gender stereotyping. These measures significantly improve inclusive membership, mutual trust, and collaborative efforts, even within limited curricula [70]. This highlights the potential for targeted interventions to create more inclusive learning environments.

Collaborations with organizations specializing in inclusivity and adaptive sports, supported by consistent funding, are essential for sustaining these initiatives. Such partnerships ensure all students benefit from school-based team sports, promoting holistic development [71,72]. Continuous evaluation of these policies and interventions is necessary to adapt to the diverse needs of students, ensuring inclusivity is effectively implemented and maintained.

## 5. Discussion

The present narrative review underscores the multifaceted advantages of school-based team sports on student wellness, revealing their profound impact across diverse dimensions such as physical, emotional, intellectual, social, spiritual, and occupational [1,2]. These insights are significant for educators, policymakers, and sports program administrators who foster holistic student development [69]. Integrating school-based team sports into educational curricula and wellness programs is pivotal to harnessing these benefits fully. Schools should incorporate team sports into core curricula through mandatory physical education and extensive extracurricular programs to ensure wide-reaching participation [23,24,73].

To maximize the benefits, it is essential to prioritize ongoing professional development for coaches. This training should emphasize inclusive practices, mental health awareness, and holistic student development [59,74]. Coaches equipped with these skills can create a more supportive and inclusive environment that addresses the diverse needs of all students. Additionally, fostering partnerships with local sports clubs, health organizations, and community groups can provide schools with the necessary resources and expertise to implement comprehensive programs [34,35,66,75]. These collaborations can help address logistical and financial constraints, making team sports more accessible to all students. Policymakers are crucial in promoting gender equality, inclusivity, and accessibility in school-based team sports. They should develop and enforce policies that ensure adequate funding for facilities and programs, especially for under-represented groups [58,59,60,76]. Such policies can help bridge the gap between different socioeconomic backgrounds and provide equal opportunities for all students to participate in team sports.

Building on the findings of this narrative review, it is evident that future research is needed to address several gaps to enhance our understanding and application of the benefits of school-based team sports. One pressing need is to explore strategies to make these sports more inclusive and accessible, particularly for students from lower socioeconomic backgrounds, those with disabilities, and culturally diverse populations. Research should focus on identifying and removing barriers to participation and creating supportive environments that encourage involvement from all students. Additionally, most existing studies are cross-sectional, providing only a snapshot of benefits at a single point in time. Longitudinal studies are needed to examine the long-term impacts of school-based team sports on various wellness dimensions, offering more profound insights into how sustained participation influences lifelong health and well-being. These studies can provide valuable data on how team sports contribute to developing skills and attributes that persist into adulthood, such as teamwork, leadership, and resilience. Furthermore, research should develop and test specific intervention strategies to enhance the benefits of team sports. For instance, integrating mental health support, leadership training, and academic tutoring within sports programs can amplify their holistic impact. Programs that combine physical activity with educational and emotional support can foster a more well-rounded development of students.

Although digital technologies and AI do not directly relate to this manuscript’s main theme, their potential impact cannot be overlooked. With the advent of digital technologies and artificial intelligence, future research should explore how these innovations can be leveraged to enhance school-based team sports. For instance, AI can personalize training programs by providing tailored feedback and recommendations based on individual student performance, thus ensuring that each student receives optimal guidance for their development [77]. Additionally, virtual reality can simulate team scenarios, offering novel and immersive ways to engage students and optimize both their physical and cognitive development. These technologies can make training more interactive and engaging, potentially increasing student participation and retention in sports programs [78]. By incorporating digital technologies and AI, school-based team sports can be more inclusive and adaptable, catering to diverse student needs and promoting holistic wellness more effectively [79].

Integrating school-based team sports into educational curricula and wellness programs substantially benefits student wellness. To fully realize these benefits, it is imperative to focus on inclusivity, accessibility, and innovative technologies. By addressing these areas through research and policy development, we can create a more equitable and effective system that fosters the holistic development of all students, preparing them to navigate the complexities of modern life with resilience and confidence.

## 6. Conclusions

School-based team sports significantly enhance student wellness across multiple dimensions, including physical health, emotional stability, intellectual growth, social skills, spiritual well-being, and career readiness. These activities promote fitness, teamwork, and social interaction, contributing to overall student development and success. Integrating team sports into school curricula and wellness programs offers numerous benefits, from improved health and reduced anxiety to better academic performance and stronger community bonds. To ensure all students benefit, it is vital to address barriers such as economic disparities, gender inequalities, and accessibility for students with disabilities. Future research should focus on long-term impacts and inclusive strategies, leveraging digital technologies for personalized training. By prioritizing school-based team sports, educational institutions can develop well-rounded, resilient, and socially responsible students ready for modern life’s challenges.

## Figures and Tables

**Figure 1 behavsci-14-00528-f001:**
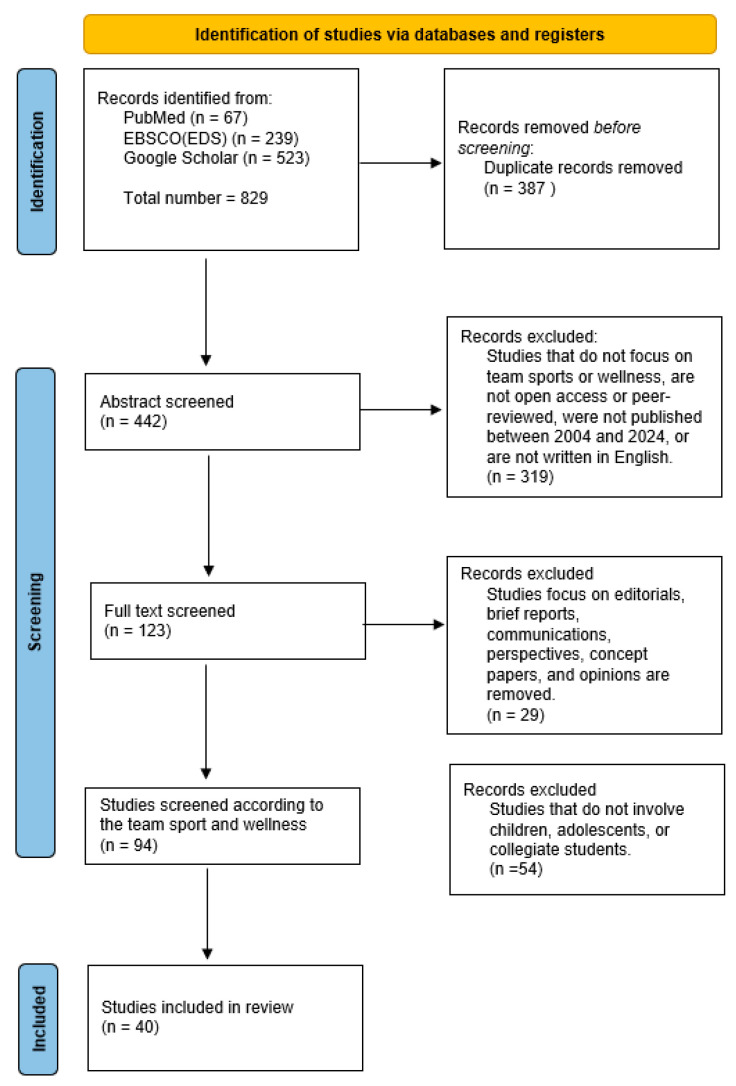
Flow diagram and selection of original articles.

**Table 1 behavsci-14-00528-t001:** Overview of physical wellness: enhancing bodily strength via team activities.

Aspect of Physical Wellness	Key Findings	References: Grade ^1^
Cardiovascular Fitness	Enhanced through aerobic and anaerobic activities	[15]: A
Muscular Strength	Improved via specific anaerobic training routines	[16]: C, [17]: A
Body Composition	Improved body composition with decreased fat mass and increased lean muscle mass	[15]: A, [18]: D, [19]: C
Health Indicators	Improved blood pressure, lipid profiles, and glucose control	[15]: A, [18]: D, [19]: C
Chronic Disease Prevention	Reduces risks of heart disease, diabetes, and obesity	[15]: A, [18]: D, [19]: C
Sustained Physical Activity	Promotes sustained physical activity through social interaction	[20]: C
Overall Physical Wellness	Overall improvements in running ability, muscle strength, heart function, and flexibility	[21]: B, [22]: D, [23]: B, [24]: B
Enjoyment	Encourages long-term engagement and sustained participation in physical activity through enjoyment and diverse challenges	[25]: B

^1^ Each reference’s evidence level is graded as A: systematic reviews and meta-analyses; B: randomized controlled trials (RCTs); C: cohort studies, case-control studies, cross-sectional surveys, case studies, and/or observational studies; D: review or evidence insufficient for categories A to C.

**Table 2 behavsci-14-00528-t002:** Summary of emotional wellness benefits through school-based team sports participation.

Aspect of Emotional Wellness	Key Findings	References: Grade ^1^
Reduction in Depression, Anxiety, and Stress	Significant mental health benefits through social support systems	[26]: A, [27]: C
Improved Self-Esteem and Social Interaction	Team sports more effective than individual activities	[26]: A, [27]: C
Higher Life Satisfaction	Correlates with lower depressive symptoms and higher life satisfaction	[26]: A, [28]: C, [29]: C
Development of Emotional Competencies	Fosters empathy, emotional regulation, and resilience	[30]: C
Reduction in Social Anxiety and Loneliness	Particularly beneficial for adolescents with greater pre-existing psychological difficulties	[31]: C, [32]: C, [33]: C
Improved Cognitive Function and Emotion Regulation	Vital social and psychological support improves cognitive function and emotion regulation	[34]: B, [35]: D
Enhanced Self-Worth, Commitment, and Effort	Strong group identification enhances self-worth and commitment	[36]: D, [37]: C

^1^ Each reference’s evidence level is graded as A: systematic reviews and meta-analyses; B: randomized controlled trials (RCTs); C: cohort studies, case-control studies, cross-sectional surveys, case studies, and/or observational studies; D: review or evidence insufficient for categories A to C.

**Table 3 behavsci-14-00528-t003:** Summary of intellectual wellness enhancement through school-based team sports.

Aspect of Intellectual Wellness	Key Findings	References: Grade ^1^
Academic Performance	Positively impacted by sports participation	[38]: D
Complex Motor Skills	Directly linked to academic improvement	[38]: D
Cognitive Performance	Positive effects, especially for boys	[39]: D
GPA Improvement	Higher GPAs associated with sports team participation	[40]: C
Dual Career Benefits	Young athletes perceive more benefits than harms	[41]: C

^1^ Each reference’s evidence level is graded as C: cohort studies, case-control studies, cross-sectional surveys, case studies, and/or observational studies; D: review or evidence insufficient for categories C.

**Table 4 behavsci-14-00528-t004:** Summary of social wellness benefits through school-based team sports participation.

Aspect of Social Wellness	Key Findings	References: Grade ^1^
Development of Social Skills	Structured environment for social skills development	[42]: A
Building Relationships	Enhances relationship-building and community	[43]: D
Fostering Community	Promotes a sense of belonging and community	[43]: D
Enhancing Cognitive and Social Life Skills	Enhances cognitive and social life skills	[42]: A
Improvements in Sport-Skills and Life-Skills Knowledge	Improves sport skills tests and life skills knowledge	[43]: D
Positive Team Environment	Retains participants by fostering enjoyment and social identity	[44]: C
Addressing Broader Social Issues	Addresses social issues through intentional program design	[45]: D
Greater School Involvement and Peer Acceptance	Reduces risky behaviors and promotes school involvement	[42]: A, [44]: C, [45]: D
Social Cohesion and Community Integration	Bridges cultural gaps and enhances social assimilation	[46]: C, [47]: D

^1^ Each reference’s evidence level is graded as A: systematic reviews and meta-analyses; C: cohort studies, case-control studies, cross-sectional surveys, case studies, and/or observational studies; D: review or evidence insufficient for categories A and C.

**Table 5 behavsci-14-00528-t005:** Summary of spiritual wellness enhancement through school-based team sports participation.

Aspect of Spiritual Wellness	Key Findings	References: Grade ^1^
Reduction in Depression and Suicidal Ideation	Sports participation reduced depression across all groups and suicidal ideation except in LGBQ youth	[48]: D
Personal and Spiritual Growth	Linked to pathways to self-transcendence and excellence	[49]: C
Higher Levels of Spirituality	Higher spirituality levels with variations by gender and ethnicity	[49]: C
Enhancement of Sporting Performance	Spirituality enhances sporting performance, personal growth, and well-being	[50]: D
Incorporation in Sports Psychology	Advocated to be included in sports psychology training	[50]: D
Influence on Treatment Progress	Athletic trainers should support athletes’ spiritual needs	[51]: C

^1^ Each reference’s evidence level is graded as C: cohort studies, case-control studies, cross-sectional surveys, case studies, and/or observational studies; D: review or evidence insufficient for categories C.

**Table 6 behavsci-14-00528-t006:** Summary of occupational wellness benefits through school-based team sports participation.

Aspect of Occupational Wellness	Key Findings	References: Grade ^1^
Development of Life Skills	Fosters essential life skills	[52]: D, [53]: C
Teamwork	Enhanced through team participation	[54]: C
Leadership	Promotes leadership abilities	[54]: C
Problem-Solving	Improves problem-solving skills	[54]: C
Entrepreneurship	Develops entrepreneurial competencies	[52]: D, [53]: C
Time Management	Higher scores in time management	[54]: C
Professional Competencies	Prepares for future professional challenges	[54]: C

^1^ Each reference’s evidence level is graded as C: cohort studies, case-control studies, cross-sectional surveys, case studies, and/or observational studies; D: review or evidence insufficient for categories C.

## Data Availability

Not applicable.

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
