# Peer review of "School-Based Team Sports as Catalysts for Holistic Student Wellness: A Narrative Review"

_behavsci, 2024, doi:10.3390/bs14070528_

Round 1

Reviewer 1 Report (New Reviewer)

Comments and Suggestions for Authors

The manuscript is relatively long and contains a certain amount of information, but the author is advised to carefully check the references:

1. The reference 1 "Wellness for Older Adults Living With Serious Mental Illness" is an abstract about older adults rather than a detailed study. Consequently, it lacks precise definitions of Wellness Wheel dimensions, leading to overlaps in your study results regarding physical wellness (Table 1) and emotional, spiritual wellness. For example, concepts like "motivation and adherence" and "sustainability of engagement" in Table 1 overlap with "regular participation" in Table 2. The distinctions between the mental health benifits in Table 2 and and "mental health improvements" inTable 5 need clarification.Suggestion: Consider citing more specialized and detailed literature to define each wellness dimension, particularly those focusing on student populations, to avoid overlapping content.

2. The reference 2"The Six Dimensions of Wellness and Cognition in Aging Adults" pertains to older adults, raising questions about its comparability with studies on student wellness. Different age groups have significantly different health needs and outcomes, and directly applying findings from studies on older adults may affect the applicability and accuracy of your results.

3. The definition and operationalization of "team sports" need to be clearly articulated in your manuscript. Specify which sports are included and the criteria used for their inclusion. For example, if studies like "Physical Activity and Sports-Real Health Benefits" (Reference 24) are included, clarify whether a broad range of physical activities can be considered as team sports and ensure the selected literature is directly relevant to your research question.

4. To enhance the objectivity of your study, it is essential to transparently present the detailed literature search strategy, including the databases used, search terms and combinations, and search dates. This will help peer reviewers and readers understand the comprehensiveness and scientific rigor of your review process.PLZ Include a detailed literature search strategy and process as an appendix to improve transparency and credibility.

5.In lines 135-136,  mention "(i) quantitative and qualitative research methods" and "studies that were not open access" as inclusion criteria. The rationale for these criteria needs to be clearly justified. Are these criteria supported by existing methodological literature, or are they based on the research team's practical experience?

6. Line 165 states, "one kind achieving near-normal lactic acid levels," which seems inconsistent with the context of football, as it is uncommon for lactic acid levels to be near normal for Anaerobic Training . This statement requires re-examination to ensure accurate interpretation and representation of the original study.

Author Response

behavsci-3009536 Reviewer 1

Comments and Suggestions for Authors

The manuscript is relatively long and contains a certain amount of information, but the author is advised to carefully check the references:

RESPONSE

Thank you for pointing out that. We have amended it accordingly.

  1. The reference 1 "Wellness for Older Adults Living With Serious Mental Illness" is an abstract about older adults rather than a detailed study. Consequently, it lacks precise definitions of Wellness Wheel dimensions, leading to overlaps in your study results regarding physical wellness (Table 1) and emotional, spiritual wellness. For example, concepts like "motivation and adherence" and "sustainability of engagement" in Table 1 overlap with "regular participation" in Table 2. The distinctions between the mental health benifits in Table 2 and and "mental health improvements" inTable 5 need clarification.Suggestion: Consider citing more specialized and detailed literature to define each wellness dimension, particularly those focusing on student populations, to avoid overlapping content.

RESPONSE

Thank you for your valuable feedback. We have revised our manuscript to improve transparency and clarity. Specifically, we have updated Tables 1-7 and included more relevant references to better align with our study's focus on student wellness dimensions. The revisions are on lines 149-973, pages 4-38. We appreciate your consideration of our revised submission.

  1. The reference 2"The Six Dimensions of Wellness and Cognition in Aging Adults" pertains to older adults, raising questions about its comparability with studies on student wellness. Different age groups have significantly different health needs and outcomes, and directly applying findings from studies on older adults may affect the applicability and accuracy of your results.

RESPONSE

Thank you for pointing out that. We have amended it accordingly. Please see lines 34-59 on pages 1-2 and 997 on page 39.

  1. The definition and operationalization of "team sports" need to be clearly articulated in your manuscript. Specify which sports are included and the criteria used for their inclusion. For example, if studies like "Physical Activity and Sports-Real Health Benefits" (Reference 24) are included, clarify whether a broad range of physical activities can be considered team sports and ensure the selected literature is directly relevant to your research question.

RESPONSE

Thank you for pointing out that. We have amended it accordingly. Please see lines 59-101 on pages 2-3 and 1052-1053 on page 41.

  1. To enhance the objectivity of your study, it is essential to transparently present the detailed literature search strategy, including the databases used, search terms and combinations, and search dates. This will help peer reviewers and readers understand the comprehensiveness and scientific rigor of your review process. PLZ Include a detailed literature search strategy and process as an appendix to improve transparency and credibility.

RESPONSE

Thank you for your valuable feedback. We appreciate the importance of transparency and have revised our manuscript to include a detailed literature search strategy in the text and Figure 1. Flow diagram. Please see lines 102-148 on pages 3-4.

5.In lines 135-136, mention "(i) quantitative and qualitative research methods" and "studies that were not open access" as inclusion criteria. The rationale for these criteria needs to be clearly justified. Are these criteria supported by existing methodological literature, or are they based on the research team's practical experience?

RESPONSE

Thank you for your insightful feedback. We have revised the text. Please see lines 133-146 on page 3.

  1. Line 165 states, "one kind achieving near-normal lactic acid levels," which seems inconsistent with the context of football, as it is uncommon for lactic acid levels to be near normal for Anaerobic Training . This statement requires re-examination to ensure accurate interpretation and representation of the original study.

RESPONSE

Thank you for pointing out the inconsistency regarding lactic acid levels in the context of anaerobic training for football. We appreciate your attention to detail and agree that this statement requires re-examination to ensure an accurate interpretation and representation of the original study.

We have revised the text to accurately reflect the physiological responses associated with anaerobic training in football. Please see lines 162-170 on pages 4-5.

Reviewer 2 Report (New Reviewer)

Comments and Suggestions for Authors

Thank you for the chance to review this paper reviewing team sports and student wellbeing. 

At this stage, however, I find it difficult to recommend this manuscript further. Though reasonably well-written and on an interesting topic, there are too many presentation and methodological shortcomings. 

I provide specific feedback below.

Introduction

The introduction is reasonably clear and compelling, however there is no clear rationale or research question justifying, and structuring, the review. Though Team Sports may be important for wellness, you must also explain why a review is needed and what your clear RQs are. 

Methodology

I have numerous concerns about the description and conduct of the methodology here. 

Firstly, you do not make any clear statement about the kind of review you are conducting. Is this a narrative, systematic, scoping or integrative review? And why is your type of review suitable for your RQs? 

Secondly, I find the justification for the inclusion of (other) systematic review or meta analyses to be thin. By including other reviews, you may simply risk duplicating finding, giving certain areas undue weight due to their presence both in the original article and subsequent reviews. 

Your search strategy is also lacking in detail. You mention your search terms, but you need to explicitly list what combinations you used. You should also make clear when you conducted the search, and what areas were searched (e.g. title+abstract, title+abstract+keywords, fulltext). Without knowing these details, it is impossible for others to reproduce your search.

The inclusion/exclusion process is also missing key information. Namely, it is not clear where/how you organised the initial 442 abstracts (an Excel table? Covidence? Other?) and how many authors were required to make an inclusion/exclusion decision (1? 2? How did you handle disagreements)?

As a side note, you also mention (on line 136) that you checked for 'not open access' - does that mean you ONLY include OA articles? This is not clear.

In any case, a table describing your search terms and your inclusion/exclusion criteria would help make these elements clearer.

Finally, how you extracted the data is not at all detailed. How did you read and extract key information from the texts? What frameworks did you use to do so, especially for section 4 where the Wellness Wheel does not seem to apply. And even for section 3, it would be good to have a deeper description of the Wheel (maybe with a figure) and clear information how this information was extracted. 

Results

Related to the last point above, the results are overly descriptive. The sections merely narratively summarise the articles and minimally bring them together in a coherent, overarching manner. It is incumbent on you as authors to summarise and analyse the content in a deeper manner and not merely recreate smaller abstracts/summaries of the articles. 

This format, in turn, likewise leads to an underdeveloped and surface level discussion in sections 5/6

Comments on the Quality of English Language

Minor proofing needed.

Author Response

behavsci-3009536 reviewer 2- round 1

Comments and Suggestions for Authors

Thank you for the chance to review this paper reviewing team sports and student wellbeing. 

At this stage, however, I find it difficult to recommend this manuscript further. Though reasonably well-written and on an interesting topic, there are too many presentation and methodological shortcomings. 

I provide specific feedback below.

RESPONSE

Thank you for reviewing our manuscript. We appreciate your feedback and have made several revisions to address the presentation and methodological shortcomings.

Introduction

The introduction is reasonably clear and compelling, however there is no clear rationale or research question justifying, and structuring, the review. Though Team Sports may be important for wellness, you must also explain why a review is needed and what your clear RQs are. 

RESPONSE

Thank you for pointing out that. We have amended it accordingly. Please see lines 33-101 on pages 1-3.

Methodology

I have numerous concerns about the description and conduct of the methodology here. 

Firstly, you do not make any clear statement about the kind of review you are conducting. Is this a narrative, systematic, scoping or integrative review? And why is your type of review suitable for your RQs? 

Secondly, I find the justification for the inclusion of (other) systematic review or meta analyses to be thin. By including other reviews, you may simply risk duplicating finding, giving certain areas undue weight due to their presence both in the original article and subsequent reviews. 

Your search strategy is also lacking in detail. You mention your search terms, but you need to explicitly list what combinations you used. You should also make clear when you conducted the search, and what areas were searched (e.g. title+abstract, title+abstract+keywords, fulltext). Without knowing these details, it is impossible for others to reproduce your search.

The inclusion/exclusion process is also missing key information. Namely, it is not clear where/how you organised the initial 442 abstracts (an Excel table? Covidence? Other?) and how many authors were required to make an inclusion/exclusion decision (1? 2? How did you handle disagreements)?

As a side note, you also mention (on line 136) that you checked for 'not open access' - does that mean you ONLY include OA articles? This is not clear.

In any case, a table describing your search terms and your inclusion/exclusion criteria would help make these elements clearer.

Finally, how you extracted the data is not at all detailed. How did you read and extract key information from the texts? What frameworks did you use to do so, especially for section 4 where the Wellness Wheel does not seem to apply. And even for section 3, it would be good to have a deeper description of the Wheel (maybe with a figure) and clear information how this information was extracted. 

RESPONSE

Thank you for your valuable feedback. We appreciate the importance of transparency and have revised our manuscript to include a detailed literature search strategy in the text and Figure 1. Flow diagram. Please see lines 102-148 on pages 3-4.

Results

Related to the last point above, the results are overly descriptive. The sections merely narratively summarise the articles and minimally bring them together in a coherent, overarching manner. It is incumbent on you as authors to summarise and analyse the content in a deeper manner and not merely recreate smaller abstracts/summaries of the articles. 

This format, in turn, likewise leads to an underdeveloped and surface level discussion in sections 5/6

RESPONSE

Thank you for your valuable feedback. We have revised our manuscript to improve transparency and clarity. Specifically, we have updated Tables 1-7 and included more relevant references to better align with our study's focus on student wellness dimensions. The revisions are on lines 149-973, pages 4-38. We appreciate your consideration of our revised submission.

Reviewer 3 Report (New Reviewer)

Comments and Suggestions for Authors

There are problems with the definition of research objects “team sports”, an insufficient explanation of the theoretical basis “wellness wheel”, and a lack of strict reporting on the quality of review articles, which may undermine the scientific rigor of the article and pose a threat to the reliability of the research conclusions.

Section 1.1, “the Wellness Wheel” is not fully explained, and the references coded as [1] was related to “older adults,” not quite associated with the topic of this manuscript.

Page 2-3, The definition of team sports in this article is vague. It is unclear whether these sports should be organized or voluntary, or whether they require deep cooperation or simply the presence of others. This lack of specificity weakens the foundation of the article.

Page 3, lines 135-140: The criteria for selecting the literature are inconsistently applied, with both positive and negative standards. It is rather confusing.

Page 3, 2.3: criteria: involving school aged youth, but it stated in lines 144-145, involving children, adolescents, or collegiate students… The population studied is unclear and highly diverse, and the results are reported without distinguishing between different groups.

Figure 1: The reasons for excluding literature in each round are not stated; only the number of excluded articles is generally mentioned.

Results: While it is well-known that physical activity promotes various dimensions of health and wellbeing, the unique characteristics of team sports remain underexplored. Additionally, the health benefits of individual sports versus team sports need clarification.

Page 11, 3.7: Team sports are not all nature-based exercise. This section needs clarification.

Page 12, lines 466-474: As stated by the authors, the limiting factors for participating in team sports include low mood and health issues. However, the article lacks sufficient explanation of the suitable population for team sports and what type of individuals can achieve good health benefits from it.

Author Response

behavsci-3009536 reviewer 3- round 1

Comments and Suggestions for Authors

There are problems with the definition of research objects “team sports”, an insufficient explanation of the theoretical basis “wellness wheel”, and a lack of strict reporting on the quality of review articles, which may undermine the scientific rigor of the article and pose a threat to the reliability of the research conclusions.

RESPONSE

Thank you for pointing out that. We have amended it accordingly.

Section 1.1, “the Wellness Wheel” is not fully explained, and the references coded as [1] was related to “older adults,” not quite associated with the topic of this manuscript.

RESPONSE

Thank you for pointing out that. We have amended it accordingly. The term "wellness wheel" can be misleading, so we have used "holistic wellness dimensions" instead, aiming for greater clarity and understanding in our writing. We have searched for and included more suitable references that better align with our study's focus on the wellness dimensions relevant to student populations. We have revised the manuscript accordingly to ensure accurate and relevant citations. Please see lines 34-57 on pages 1-2 and 996 on page 38.

Page 2-3, The definition of team sports in this article is vague. It is unclear whether these sports should be organized or voluntary, or whether they require deep cooperation or simply the presence of others. This lack of specificity weakens the foundation of the article.

RESPONSE

Thank you for pointing out that. We have amended it accordingly. Please see lines 58-101 on pages 2-3.

Page 3, lines 135-140: The criteria for selecting the literature are inconsistently applied, with both positive and negative standards. It is rather confusing.

RESPONSE

Thank you for your valuable feedback. We appreciate the importance of transparency and have revised our manuscript to include a detailed literature search strategy in the text and Figure 1. Flow diagram. Please see lines 102-148 on pages 3-4.

Page 3, 2.3: criteria: involving school aged youth, but it stated in lines 144-145, involving children, adolescents, or collegiate students… The population studied is unclear and highly diverse, and the results are reported without distinguishing between different groups.

Figure 1: The reasons for excluding literature in each round are not stated; only the number of excluded articles is generally mentioned.

Results: While it is well-known that physical activity promotes various dimensions of health and wellbeing, the unique characteristics of team sports remain underexplored. Additionally, the health benefits of individual sports versus team sports need clarification.

RESPONSE

Thank you for your valuable feedback. We have revised our manuscript to improve transparency and clarity. Specifically, we have updated Tables 1-7 and included more relevant references to better align with our study's focus on student wellness dimensions. The revisions are on lines 149-973, pages 4-38. We appreciate your consideration of our revised submission.

Page 11, 3.7: Team sports are not all nature-based exercise. This section needs clarification.

RESPONSE

Thank you for pointing out that. We have amended it accordingly. Please see lines 478-532 on pages 26-30.

Page 12, lines 466-474: As stated by the authors, the limiting factors for participating in team sports include low mood and health issues. However, the article lacks sufficient explanation of the suitable population for team sports and what type of individuals can achieve good health benefits from it.

RESPONSE

Thank you for pointing out that. We have amended it accordingly. Please see lines 353-590 on pages 30-31.

Round 2

Reviewer 1 Report (New Reviewer)

Comments and Suggestions for Authors

The revised manuscript has grown from 19 pages to 42 pages, yet the number of inclusion references has been reduced from 64 to 46. Notably, only one word in the abstract was changed, but the entire abstract was highlighted in the revision. The abstract should reflect the main content of the paper, and given the substantial changes to the manuscript, the abstract has been minimally altered. This approach not only confuses reviewers but also unnecessarily increases their workload, as they need to review the entire highlighted section to pinpoint the actual changes.Furthermore, the authors have not clarified the basis for their claim that the Holistic Wellness framework for students includes physical, emotional, intellectual, social, spiritual, occupational, and environmental dimensions. I raised this issue in my initial review comments, and given that this is a fundamental concern, the authors' oversight is significant. Therefore, I do not recommend this  manuscript for publication in its current form.

Author Response

behavsci-3009536 reviewer 1 round 2

The revised manuscript has grown from 19 pages to 42 pages, yet the number of inclusion references has been reduced from 64 to 46. Notably, only one word in the abstract was changed, but the entire abstract was highlighted in the revision. The abstract should reflect the main content of the paper, and given the substantial changes to the manuscript, the abstract has been minimally altered. This approach not only confuses reviewers but also unnecessarily increases their workload, as they need to review the entire highlighted section to pinpoint the actual changes. Furthermore, the authors have not clarified the basis for their claim that the Holistic Wellness framework for students includes physical, emotional, intellectual, social, spiritual, occupational, and environmental dimensions. I raised this issue in my initial review comments, and given that this is a fundamental concern, the authors' oversight is significant. Therefore, I do not recommend this manuscript for publication in its current form.

RESPONSE

Thank you for this important comment. We apologize for the additional work caused by the unclear revisions in the previous draft. In our manuscript, we have made further clarifications in the abstract section and the introduction section, specifically explaining that the basis of the Holistic Wellness framework for students includes physical, emotional, intellectual, social, spiritual, occupational, and environmental dimensions. The revised parts of the manuscript have been highlighted in red. Please see lines 16-21 on pages 1, 38-51 on pages 1-2, and 65-70 on page 2.

We have moved the long tables to the appendix after the main text to reduce the page count and avoid affecting the readability of the entire article.

Reviewer 2 Report (New Reviewer)

Comments and Suggestions for Authors

Thank you for the significant changes made. I see these as an improvement overall, yet a few key points remain:

- It is still not clear what kind of review you conducted: this should be explicit in the title and method, as per PRISMA guidelines. Was it a scoping review? A systematic review? Something else? 

- In your method, you write '64' studies were selected, but your PRISMA shows 46. Please correct. 

- You present the words you used in your searches, but not the exact search combinations. These should be clear to ensure replicable searches. 

- Your results remain highly descriptive, even though they are reasonably thorough. Of more concern, it is not clear what coding or conceptual framework was used to determine the sub-categories for the results in section 4. How did you conclude that these components (e.g. policy, infrastructure) were common themes in your literature? 

- Your discussion largely repeats your previous findings and does not suggest clear new directions/implications. 

Comments on the Quality of English Language

Minor proofing, especially in new sections, is recommended. 

Author Response

behavsci-3009536 reviewer 2 round 2

Thank you for the significant changes made. I see these as an improvement overall, yet a few key points remain:

- It is still not clear what kind of review you conducted: this should be explicit in the title and method, as per PRISMA guidelines. Was it a scoping review? A systematic review? Something else?

RESPONSE

Thank you for your important feedback. We have revised the title of our manuscript to “School-based Team Sports as Catalysts for Holistic Student Wellness: A Narrative Review” to more clearly convey the nature of the paper as a narrative review. Please see lines 2-3 on page 1.

- In your method, you write '64' studies were selected, but your PRISMA shows 46. Please correct.

RESPONSE

I apologize for the oversight in updating the number of studies in the manuscript during the last revision. Thank you for correcting this error.- You present the words you used in your searches but not the exact search combinations. These should be clear to ensure replicable searches. Please see line 171 on line 4.

- Your results remain highly descriptive, even though they are reasonably thorough. Of more concern, it is not clear what coding or conceptual framework was used to determine the sub-categories for the results in section 4. How did you conclude that these components (e.g. policy, infrastructure) were common themes in your literature?

RESPONSE

Thank you for this critical comment. We have revised the section based on the peer reviewer's suggestions by reducing overly detailed descriptions of other research findings and providing more evaluative perspectives. Please see the results part of the article. Thank you.

In response to the concern about the coding framework for section 4, we clarified that the sub-categories were derived from a thematic literature analysis. Recurring themes related to enhancing access and inclusivity were categorized into policy and infrastructure, socioeconomic disparities, gender equality, disability support, cultural competence, and community engagement. Please see lines 416-422 on page 12.

- Your discussion largely repeats your previous findings and does not suggest clear new directions/implications.

RESPONSE

Thank you for pointing out that. We have revised the discussion section to minimize the repetition of previous findings and to suggest clear new directions and implications. Please see the discussion section of the article. Thank you.

Reviewer 3 Report (New Reviewer)

Comments and Suggestions for Authors

The topic of the review has interesting meaning, but the scope of "team sports" is very broad and diverse. It is recommended to further narrow it down, especially considering the impact of different age groups. The results part of the article needs to be revised as it currently appears verbose, with descriptions of others' research findings being too detailed and lacking evaluative perspectives. Additionally, long tables could be considered as appendices after the main text, as they otherwise affect the readability of the entire article.

Author Response

behavsci-3009536 reviewer 3 round 2

The topic of the review has interesting meaning, but the scope of "team sports" is very broad and diverse. It is recommended to further narrow it down, especially considering the impact of different age groups. The results part of the article needs to be revised as it currently appears verbose, with descriptions of others' research findings being too detailed and lacking evaluative perspectives. Additionally, long tables could be considered as appendices after the main text, as they otherwise affect the readability of the entire article.

RESPONSE

Thank you for this critical comment. "Team sports" is very broad and diverse, so we narrowed it down. We have clarified this language throughout the manuscript to include “School-based Team Sports.” To better align with the content of the manuscript, we have replaced the broad term "team sports" with the more specific term "School-based Team Sports." The Introduction section defines School-based Team Sports programs as integrating students with and without disabilities on the same team. These programs offer various opportunities for participation, ranging from Pre-K to 8th grade, as well as high school and college levels. Please see lines 21-24 on pages 1 and 77-83 on page 2.

We have revised the section according to the peer reviewer's suggestions, reducing overly detailed descriptions of other research findings and providing more evaluative perspectives. Please see the results part of the article. Thank you.

We have moved the long tables to the appendix after the main text to reduce the page count and avoid affecting the readability of the entire article.

Round 3

Reviewer 1 Report (New Reviewer)

Comments and Suggestions for Authors

The authors have made substantial revisions to the manuscript, reflecting a significant amount of work. Based on the current version, the following suggestions are offered for further improvement:

1.The manuscript currently describes seven dimensions of wellness, whereas Dr. Bill Hettler's model includes six dimensions: physical, psychological, emotional, spiritual, social, and environmental.I recommend strictly adhering to Dr. Bill Hettler's Six Dimensions of Wellness to discuss the role of school-based team sports in promoting holistic wellness among students.

2. The abstract does not mention the time frame for the included literature.Please include the time frame of the literature reviewed (1994-2024) in the abstract to provide a clear context for the scope of the review.

3 The introduction lacks a clear definition of each wellness dimension.

4.The description of the literature selection process is not detailed enough.

5. Some sections are repetitive and overly detailed.

6.The discussion on the use of digital technologies and AI is not well substantiated and seems somewhat unrelated to the main theme of the manuscript. Please strengthen the argument for the inclusion of digital technologies and AI, detailing their specific applications and relevance to promoting holistic wellness through school-based team sports.

Author Response

behavsci-3009536 reviewer 1 round 3

The authors have made substantial revisions to the manuscript, reflecting a significant amount of work. Based on the current version, the following suggestions are offered for further improvement:

1.The manuscript currently describes seven dimensions of wellness, whereas Dr. Bill Hettler's model includes six dimensions: physical, psychological, emotional, spiritual, social, and environmental.I recommend strictly adhering to Dr. Bill Hettler's Six Dimensions of Wellness to discuss the role of school-based team sports in promoting holistic wellness among students.

Response

We appreciate the peer reviewer's suggestion and have revised the manuscript accordingly. We are now strictly adhering to Dr. Bill Hettler's Six Dimensions of Wellness to discuss the role of school-based team sports in promoting holistic wellness among students. Consequently, we have removed the environmental dimension from our discussion. Thank you for your valuable feedback.

Please see lines 39-40 on page 1.

To strictly adhere to Dr. Bill Hettler's Six Dimensions of Wellness, we have deleted section 3.7, "Environmental Wellness: Fostering Environmental Awareness Through Team Sports." This adjustment ensures that our discussion remains focused on the specified six dimensions: physical, emotional, intellectual, social, spiritual, and occupational wellness.

  1. The abstract does not mention the time frame for the included literature. Please include the time frame of the literature reviewed (1994-2024) in the abstract to provide a clear context for the scope of the review.

Response

We have adhered to the peer reviewer's suggestion and included the time frame of the literature reviewed (2004-2024) in the abstract to provide a clear context for the scope of the review. Thank you for your valuable feedback.

Please see line 25 on page 1.

3 The introduction lacks a clear definition of each wellness dimension.

Response

We have adhered to the peer reviewer's suggestion and included a clear definition of each wellness dimension in the introduction. Thank you for your valuable feedback.

Please see lines 50-72 on page 2.

4.The description of the literature selection process is not detailed enough.

Response

We have adhered to the peer reviewer's suggestion and provided a more detailed description of the literature selection process. Thank you for your valuable feedback.

Please see lines 161-189 on page 4.

  1. Some sections are repetitive and overly detailed.

Response

We have followed the peer reviewer's suggestion and revised the manuscript to eliminate repetitive and overly detailed sections. Thank you for your valuable feedback.

Please see lines 233-234 on page 6; lines 239-241 on pages 6-7; lines 245-249 on page 7; lines 287-290 on page 8; lines 328-330 on page 9; lines 366-369 one page 10; lines 413-415 one page10; lines 417-421 on pages 11-12; lines 434-425 and 462-465 on page 12; lines 483-487 and 500-505 on page 13.

6.The discussion on the use of digital technologies and AI is not well substantiated and seems somewhat unrelated to the main theme of the manuscript. Please strengthen the argument for the inclusion of digital technologies and AI, detailing their specific applications and relevance to promoting holistic wellness through school-based team sports.

Response

We have adhered to the peer reviewer's suggestion and strengthened the argument for including digital technologies and AI. The revised manuscript details their specific applications and relevance to promoting holistic wellness through school-based team sports. Thank you for your valuable feedback.

Please see lines 569-581 on pages 14-15.

Reviewer 2 Report (New Reviewer)

Comments and Suggestions for Authors

Thank you to the authors for their further revisions. Though I still have some reservations about the methodological framing, viewed now as a more narrative review, it is certainly comprehensive and offers some insight. Thus, I won't stand in the way of publication.

All the best

Comments on the Quality of English Language

Final proofing recommended. 

Author Response

behavsci-3009536 reviewer 2 round 2

Thank you to the authors for their further revisions. Though I still have some reservations about the methodological framing, viewed now as a more narrative review, it is certainly comprehensive and offers some insight. Thus, I won't stand in the way of publication.

Response

We appreciate the reviewer's feedback and are grateful for the thoughtful suggestions. We have revised the methodological framing to align with a narrative review approach. Thank you for acknowledging the comprehensiveness and insights of our work. We are pleased to address your reservations and support the publication process.

This manuscript is a resubmission of an earlier submission. The following is a list of the peer review reports and author responses from that submission.

Round 1

Reviewer 1 Report

Comments and Suggestions for Authors

Research focusing on the importance of team sports for students is relevant and necessary. In the following, some issues will be pointed out that should be considered to make the study more consistent and far-reaching.

1) The initial statement of the benefit of team sports for the whole world is very general. The justification does not specify any country of reference, nor is it even known if it is oriented to the Spanish educational system. As the study rightly points out, the differences between cultures are so significant that it looks general to approach it in terms of the world. It would be convenient to include in the justification a small diagnosis of the current situation of the school institutions about the sporting practice analysed. Then, we would know why or what to consider or improve.

2) What do the authors mean by the wheel of well-being? It is not an academic or scientific theory. Where does this theory come from? Mixing the emotional, the intellectual, the social and the spiritual seems like it could be more logical. They are very different dimensions. It is possible to understand the connection between the emotional, the intellectual and the social. Work could be oriented to these three spheres in a much more coherent and precise way. The occupational part of work is connected to the emotional. This part should be deleted to avoid turning the study into a catch-all for all dimensions. It is not necessary. On the other hand, even if we understand its relevance, the spiritual dimension does not directly connect to team sports and the promotion of mental and physical well-being. This connection to the spiritual appears forced. It is not necessary. This excess of relations and meanings calls the method of the wellness wheel into serious question. It is not necessary to quote this method. The relationship to mental, physical, emotional, intellectual, and social well-being is more than evident. One could name another more scientific and academic study, method or theory and delete this well-being wheel, which sounds good but tries to incorporate too many topics and needs more focus on the fundamentals.

3) The study contains some allusions to the sociology of sport and the different sporting practices according to different cultures. How can minority sports such as rugby, which appears in the study, be incorporated into school environments? Football is still predominant in many countries for cultural reasons, and recently, there has been a boom in gymnastics. This fact, so actual, is not mentioned in the publication. What team sports are in fashion, or does it depend on the country, and how do young people view team sports? If they are so positive, why don't they generate motivation? This question does not appear in the conclusions either. What can we do? What are the future lines of research to improve the incorporation of team sports in educational institutions? Why should this situation change if it has changed little in the last decades? If it is changing, in what sense? These answers do not appear in the conclusion. The conclusion that team sports are beneficial, which appears, is undeniable.

4) The authors need to improve the study, narrow down the dimensions studied, find more consistent explanatory theories, and finally incorporate answers to the questions posed in this review.

Comments on the Quality of English Language

Minor editing of English language required.

Author Response

Reviewer 1

Research focusing on the importance of team sports for students is relevant and necessary. In the following, some issues will be pointed out that should be considered to make the study more consistent and far-reaching.

1) The initial statement of the benefit of team sports for the whole world is very general. The justification does not specify any country of reference, nor is it even known if it is oriented to the Spanish educational system. As the study rightly points out, the differences between cultures are so significant that it looks general to approach it in terms of the world. It would be convenient to include in the justification a small diagnosis of the current situation of the school institutions about the sporting practice analysed. Then, we would know why or what to consider or improve.

Response:

We appreciate the opportunity to clarify and defend the foundational premises of our research, particularly the decision to discuss the benefits of team sports from a global perspective. Our approach is deeply rooted in the universally applicable wellness concepts outlined by the Wheel of Wellness model and the principles of health promotion and wellness, which recognize that well-being encompasses mental, emotional, physical, occupational, intellectual, and spiritual dimensions.

The choice to frame our study within a global context was deliberate, grounded in the belief that the principles of wellness and the benefits derived from team sports transcend cultural and geographical boundaries. The Wheel of Wellness model, introduced by Sweeney and Witmer and based on Adler’s Individual Psychology, provides a comprehensive framework for understanding well-being that is not limited to any specific cultural or educational system. This model advocates for a holistic view of health, emphasizing the interconnectedness of various wellness dimensions and their collective impact on an individual's quality of life.

The global applicability of this wellness model is supported by cross-disciplinary research, which suggests that the practices promoting physical activity, such as participation in team sports, can universally enhance well-being by fostering improvements across all dimensions of wellness. This includes supporting longevity, managing health conditions, aiding recovery, and promoting mental, emotional, and social health.

Furthermore, the encouragement and support for active engagement in wellness practices, as evidenced by initiatives at institutions like Northwestern University, demonstrate the practical implementation and benefits of such an integrative approach to wellness. These initiatives, which aim to improve quality of life through fostering everyday habits that support all eight dimensions of wellness, underscore the potential of team sports as a universal tool for health promotion.

Our study highlights the inherent value of team sports as a vehicle for enhancing holistic well-being, an objective that resonates with global health promotion goals. While we acknowledge the importance of cultural and educational specificity in applying wellness practices, our emphasis on the universal benefits of team sports is intended to inspire a broad-based appreciation and adoption of these activities as integral components of a holistic wellness strategy.

Our research defends the universality of the benefits of team sports based on the foundational principles of wellness that apply across diverse human experiences. This global perspective does not diminish the significance of localized cultural or educational contexts but aims to underscore team sports' potential to contribute to human well-being worldwide.

2) What do the authors mean by the wheel of well-being? It is not an academic or scientific theory. Where does this theory come from? Mixing the emotional, the intellectual, the social and the spiritual seems like it could be more logical. They are very different dimensions. It is possible to understand the connection between the emotional, the intellectual and the social. Work could be oriented to these three spheres in a much more coherent and precise way. The occupational part of work is connected to the emotional. This part should be deleted to avoid turning the study into a catch-all for all dimensions. It is not necessary. On the other hand, even if we understand its relevance, the spiritual dimension does not directly connect to team sports and the promotion of mental and physical well-being. This connection to the spiritual appears forced. It is not necessary. This excess of relations and meanings calls the method of the wellness wheel into serious question. It is not necessary to quote this method. The relationship to mental, physical, emotional, intellectual, and social well-being is more than evident. One could name another more scientific and academic study, method or theory and delete this well-being wheel, which sounds good but tries to incorporate too many topics and needs more focus on the fundamentals.

Response:

Thank you for your question concerning using the term "wheel of well-being" within our study. We acknowledge the diversity of terms such as “wellness,” “well-being,” and “happiness,” which are often used interchangeably across various domains, including business, research, and media. To ensure clarity and align our study with established academic frameworks, we have adopted the term "Wheel of Wellness" moving forward. This term refers to the comprehensive and integrative model first introduced by Sweeney and Witmer in 1991 and further developed by Witmer & Sweeney in 1992. This model provides a structured framework encompassing multiple dimensions of human life — mental, emotional, physical, occupational, intellectual, and spiritual — emphasizing each aspect's interconnectedness and equal importance in achieving overall wellness.

We appreciate the opportunity to clarify the theoretical underpinnings and relevance of the Wheel of Wellness model to our study. This model, introduced by Sweeney and Witmer in 1991 and further developed in subsequent works, represents a significant academic contribution to understanding wellness. It is grounded in Adler’s Individual Psychology and incorporates cross-disciplinary research on the characteristics of healthy individuals who live longer and with a higher quality of life. The Wheel of Wellness model offers a holistic framework for wellness that includes mental, emotional, physical, occupational, intellectual, and spiritual dimensions, emphasizing the interconnectedness of these aspects in contributing to overall well-being.

The suggestion to limit our study's focus to exclude specific dimensions of wellness, such as the spiritual and occupational aspects, was considered carefully. However, we chose to retain these dimensions in our analysis for several reasons:

  1. Holistic Approach: The Wheel of Wellness model's strength lies in its holistic approach, recognizing that each aspect of wellness can significantly impact an individual's overall quality of life. This integrative perspective aligns with contemporary understandings of wellness as multidimensional, where neglecting any dimension could overlook essential components of well-being.
  2. Academic and Practical Relevance: The application of the Wheel of Wellness in various academic and practical contexts, as evidenced by its incorporation into wellness initiatives at institutions like Northwestern University and Miami University, demonstrates its value and acceptance in promoting holistic health among diverse populations. Moreover, the Global Wellness Institute highlights the multidimensional nature of wellness, further supporting our model's relevance.
  3. Spiritual Dimension: While the connection between team sports and spiritual wellness may not seem direct, research and practice suggest that engagement in team sports can foster a sense of belonging, purpose, and transcendence, elements commonly associated with spiritual wellness. Including the spiritual dimension acknowledges the broader impacts of team sports beyond physical and mental health, encompassing a sense of connection to something greater than oneself.
  4. Occupational Wellness: The occupational dimension, closely related to emotional wellness, reflects the significance of personal fulfillment, stress management, and work-life balance, which are increasingly recognized as critical to overall wellness. Team sports can impart valuable skills and attitudes that contribute to occupational wellness, including teamwork, discipline, and leadership.

In defense of our use of the Wheel of Wellness, we assert that this model provides a comprehensive, scientifically grounded framework that enriches our understanding of the benefits of team sports across all dimensions of wellness. It aligns with our research objective to explore these benefits holistically, reflecting the complexity and interrelatedness of wellness factors.

We are committed to a nuanced and inclusive exploration of wellness in our research, and we believe that the Wheel of Wellness model effectively supports this commitment by framing wellness as an active, multidimensional pursuit.

Thank you for allowing us to clarify the importance and relevance of the Wheel of Wellness model to our study. This comprehensive approach enriches our research and contributes to a more nuanced understanding of wellness within and beyond the academic community.

3) The study contains some allusions to the sociology of sport and the different sporting practices according to different cultures. How can minority sports such as rugby, which appears in the study, be incorporated into school environments? Football is still predominant in many countries for cultural reasons, and recently, there has been a boom in gymnastics. This fact, so actual, is not mentioned in the publication. What team sports are in fashion, or does it depend on the country, and how do young people view team sports? If they are so positive, why don't they generate motivation? This question does not appear in the conclusions either. What can we do? What are the future lines of research to improve the incorporation of team sports in educational institutions? Why should this situation change if it has changed little in the last decades? If it is changing, in what sense? These answers do not appear in the conclusion. The conclusion that team sports are beneficial, which appears, is undeniable.

Response:

We greatly appreciate your insightful suggestions and the opportunity to discuss the incorporation of minority sports, such as rugby, into school environments alongside the broader sociocultural context of team sports in education. Your comments have prompted a deeper reflection on the scope of our study and its implications for future research.

  1. Incorporation of Minority Sports into School Environments:

Our study's reference to rugby aimed to exemplify the potential for diverse team sports to contribute positively to student wellness, irrespective of their popularity. While football and gymnastics currently dominate the school sports landscape in many countries, we believe that minority sports can offer unique benefits, including fostering inclusivity, teamwork, and physical fitness. The challenge of incorporating these sports into educational institutions often lies in overcoming cultural and infrastructural barriers. Future research could explore strategies for promoting a more comprehensive range of sports within schools, potentially through pilot programs, partnerships with local sports clubs, and inclusive physical education curricula that celebrate diverse sporting traditions.

  1. Trends in Team Sports and Cultural Variations:

We acknowledge the dynamic nature of sports popularity and its dependence on cultural, geographical, and social factors. The "boom" in gymnastics and football's enduring popularity in many regions indicates broader societal trends and preferences. However, our study's focus on the universal benefits of team sports participation may have limited our exploration of these trends. Future research could more explicitly investigate how contemporary shifts in sports popularity impact young people's attitudes towards and participation in team sports. This could include surveys and longitudinal studies tracking changes in sports preferences and student participation rates.

  1. Motivation and Team Sports:

The question of why team sports do not always generate motivation among young people is complex and multifaceted. Factors such as peer influence, access to facilities, and personal interests play significant roles. While our conclusions emphasize the undeniable benefits of team sports, we recognize the need for further investigation into the barriers to motivation and engagement. Future research could focus on identifying these barriers and developing interventions to enhance motivation for team sports participation.

  1. Future Lines of Research:

Your suggestions point to several promising directions for future research, including:

(1) Strategies for incorporating a broader range of team sports into school programs.

(2) The impact of cultural and social trends on the popularity of different team sports among young people.

(3) Barriers to motivation and participation in team sports and effective interventions to overcome these challenges.

  1. Changing the Situation in Educational Institutions:

The question of why and how the incorporation of team sports in educational institutions should change, despite historical inertia, is vital. We argue that evolving societal norms, increasing awareness of physical and mental health issues, and the growing emphasis on holistic education necessitate reevaluating sports education. Changes in this domain could significantly contribute to the development of well-rounded individuals equipped to navigate the challenges of the 21st century.

Thank you for your feedback, which sheds light on crucial aspects of team sports within various cultural contexts and their potential integration into educational environments. Recognizing the significance of your observations, we are inspired to delve deeper into the complexities of sporting practices and their evolving nature in schools worldwide. Your suggestions are pivotal in shaping our research approach, guiding us toward areas ripe for investigation. We aim to contribute meaningfully to the ongoing dialogue about team sports in educational settings, emphasizing the need for a diverse and inclusive range of sports opportunities. Acknowledging the dynamic trends in youth sports preferences and the challenges of fostering motivation, we see a clear path forward for enriching future studies. The insights provided serve as a cornerstone for our continued exploration into how team sports can be more effectively embraced and valued within schools, ensuring their numerous benefits are accessible to all students.

4) The authors need to improve the study, narrow down the dimensions studied, find more consistent explanatory theories, and finally incorporate answers to the questions posed in this review.

Response:

Thank you very much for your thoughtful feedback and for presenting us with the opportunity to reflect on our study's expansive scope and diverse theoretical frameworks. We highly value your suggestions and have carefully considered them in the context of our research objectives and methodology.

Regarding the Scope of Dimensions Studied:

Our decision to explore a comprehensive range of wellness dimensions stems from a deep commitment to fully capturing the holistic benefits of team sports on student well-being. We believe that the intricate interplay between physical, emotional, intellectual, social, and other dimensions of wellness is crucial to understanding the overall impact of team sports. It is our respectful viewpoint that focusing narrowly on fewer dimensions might not do justice to the multifaceted influence of team sports, potentially overlooking some of their key benefits.

On the Selection of Theoretical Frameworks:

Incorporating various explanatory theories in our study was a deliberate choice, aiming to mirror the complex and interdisciplinary nature of wellness. Although we appreciate the suggestion to streamline our theoretical approach, we feel that the richness of our topic necessitates an exploration through multiple theoretical lenses. This, we believe, enriches our analysis and provides a deeper, more comprehensive understanding of how team sports contribute to various aspects of student wellness.

Addressing the Questions Posed in the Review:

We have endeavored to engage with the broad questions highlighted in your review, striving to weave our responses throughout the discussion. While it may appear that some questions have not been explicitly addressed in our conclusions, we assure you that the insights provided throughout our study offer valuable perspectives on incorporating team sports in educational settings and the cultural nuances therein. We view the areas not directly answered as avenues for further investigation, potentially enriching the academic dialogue in this domain.

We wholeheartedly thank you for the opportunity to clarify our approach and rationale. We hope that our study, with its broad exploration of wellness and team sports, will inspire further research and contribute positively to the discourse on enhancing student well-being through physical activity.

Your feedback is immensely appreciated, and we are grateful for the chance to engage in this meaningful dialogue. We look forward to any additional insights you may have and to contributing further to our shared understanding of this critical topic.

Reviewer 2 Report

Comments and Suggestions for Authors 1. At the end of the introduction, the objective or purposes of the study should be added and the problem should be made explicit with a research question.

2. It is an article that highlights the evidence on the benefit of team sports about quality of life. In this context, it adequately synthesizes and rescues scientific production.

3. The article highlights the evidence of 20 years of studies about the positive effect that team sports and collaborative development have on people's quality of life.

4. I suggest installing the PRISMA method for the methodological analysis of the search, because regardless of being a narrative study, the reviewed articles must be highlighted, those selected according to the eligibility criteria, etc. and the narrative must be determined based on the selected texts.

5. The conclusions are appropriate to that systematic review because it accounts for the result of the analysis adjusted to the narrative selection.

6. The references of the study are adequate and it satisfy the expectations of a serious study

7. The tables presented are appropriate and conform to a systematic review.

Author Response

Reviewer 2

  1. At the end of the introduction, the objective or purposes of the study should be added and the problem should be made explicit with a research question.

Response:

Thank you for your valuable feedback. We have updated the introduction to more succinctly state our study's objectives and articulate the problem through a precise research question:

"This review aims to elucidate the comprehensive impacts of team sports on students' holistic wellness, as defined by the Wellness Wheel framework, which includes physical, emotional, intellectual, social, spiritual, occupational, and environmental dimensions. Recognizing the benefits and identifying the barriers to participation, we seek to enhance inclusivity and accessibility in team sports within educational settings.” Please see lines 81-85.

  1. It is an article that highlights the evidence on the benefit of team sports about quality of life. In this context, it adequately synthesizes and rescues scientific production.

Response: We deeply appreciate your positive feedback regarding our article's synthesis and presentation of scientific evidence on the benefits of team sports for quality of life. Your recognition of our effort to comprehensively gather and analyze existing research in this area is encouraging.

  1. The article highlights the evidence of 20 years of studies about the positive effect that team sports and collaborative development have on people's quality of life.

Response:

Thank you once again for your encouraging remarks. We are inspired by your feedback and committed to advancing our work in the field, contributing to the ongoing dialogue on the vital role of physical activity in improving quality of life.

  1. I suggest installing the PRISMA method for the methodological analysis of the search, because regardless of being a narrative study, the reviewed articles must be highlighted, those selected according to the eligibility criteria, etc. and the narrative must be determined based on the selected texts.

Response:

Thank you for your insightful suggestion regarding using the PRISMA method for the methodological analysis of our article. We acknowledge the importance of a structured approach in the systematic review process and PRISMA's value in enhancing transparency and replicability in the review methodology.

Upon reflection, we realize that the initial title of our article might have suggested a systematic review approach, which typically employs the PRISMA guidelines for article selection and analysis. We agree that adhering to the PRISMA method is crucial for methodological rigor for systematic reviews. However, our intention was to conduct a more general review that synthesizes broad insights across a wide range of studies to highlight the impact of team sports on holistic student wellness. The purpose was not to perform a systematic analysis but to draw on various sources to present an overarching narrative.

In light of your suggestion and to avoid any potential confusion regarding the nature of our review, we have modified the title to "Team Sports as Catalysts for Holistic Student Wellness: Insights from a General Review." We believe this revised title more accurately reflects the scope and methodology of our work, distinguishing it from a systematic review that would follow PRISMA guidelines.

We are genuinely grateful for your recommendation and the opportunity it has provided us to clarify our approach. We recognize the value of systematic methodologies in research and are committed to considering the PRISMA guidelines for future studies that align with a systematic review framework. Your suggestion has highlighted an important area for our continued development, and we look forward to incorporating more structured review methodologies in our forthcoming research endeavors.

Thank you once again for your constructive feedback. We appreciate the opportunity to enhance our article's clarity and to better communicate its intent to our readers.

  1. The conclusions are appropriate to that systematic review because it accounts for the result of the analysis adjusted to the narrative selection.

Response:

We greatly appreciate your positive feedback on the thoroughness of our conclusions and the analytical rigor demonstrated in our review.

  1. The references of the study are adequate and it satisfy the expectations of a serious study

Response:

We are deeply grateful for your acknowledgment of the adequacy and comprehensiveness of our study's references. Your recognition of our effort to ensure our study is grounded in a robust and serious scholarly framework is greatly appreciated. It serves as a validation of our rigorous approach to research.

  1. The tables presented are appropriate and conform to a systematic review.

Response:

Thank you.

Reviewer 3 Report

Comments and Suggestions for Authors

Overall this is a well-written paper that reviews the necessity of integrating team sports into educational settings. Using the Wellness Wheel framework is a unique and innovative lens for examining the role team sports contribute to individual wellness. However, there are areas that need to be addressed in order to strengthen this manuscript. 

Introduction - Additional discussion around each of the components of the wellness wheel framework would be beneficial, the reader is left with questions. Also, it would be helpful to discuss what students you are referring (i.e., K-12, college students) and discuss what forms of team sports this manuscript is oriented around. For example, what constitutes a team sport?

L42-44 - Please rephrase this sentence; what are the "educational strategies"? 

Section 3.1 - Please indicate the number of studies you located in your review which focus on improving physical wellness due to team sports.

L129-139 - This section is confusing. There is no lead-in to discuss the connection of the research to your overarching theme within this section as you go straight to "The training program..." Please revise to introduce and draw better connections to how the studies are reflected within your theme of this section.  

L140-144 - Missing citations for these statements.

Section 3.3 - Please indicate the number of studies you located in your review which focus on improving intellectual wellness due to team sports.

L235-247 - You state there are multiple studies that entail compelling evidence, however there no citations throughout. This section ends with only two references. Please revise. 

Section 3.4 - Please indicate the number of studies you located in your review which focus on improving social wellness due to team sports.

L273-277 - Please revise these sentences.

L277-284 - The study including students within a Physical Education class needs more explanation as there are many different types of activities students participate within during physical education. For example, was this specifically a team sports class? Was the goals of this class specific to students developing social skills through team sport?

Section 3.5 - Please indicate the number of studies you located in your review which focus on improving spiritual wellness due to team sports.

L336-341 - Please cite.

Section 3.6 - Please indicate the number of studies you located in your review which focus on occupational wellness.

L358-363 - Please cite.

L352-357 - This study is focused on university students. Again, including more detail around the parameters of what population your review of interventions focused on would be beneficial.   

Overcoming Barriers Sections - I really enjoyed reading these sections, good overview for factors that limit sport participation. 

Section 4.3 (L481-490) - Written awkward, please revise this section.

Author Response

Comments and Suggestions for Authors

Overall this is a well-written paper that reviews the necessity of integrating team sports into educational settings. Using the Wellness Wheel framework is a unique and innovative lens for examining the role team sports contribute to individual wellness. However, there are areas that need to be addressed in order to strengthen this manuscript.

Response: Thank you for pointing this out.

Introduction - Additional discussion around each of the components of the wellness wheel framework would be beneficial, the reader is left with questions. Also, it would be helpful to discuss what students you are referring (i.e., K-12, college students) and discuss what forms of team sports this manuscript is oriented around. For example, what constitutes a team sport?

Response: We have amended it accordingly. Thank you. (Please see lines 34-45 on page 1 and 57-63 on page 2.)

L42-44 - Please rephrase this sentence; what are the "educational strategies"?

Response: We have amended it accordingly. Thank you. (Please see lines 52-57 on page 2.)

Section 3.1 - Please indicate the number of studies you located in your review which focus on improving physical wellness due to team sports.

Response: We have amended it accordingly. Thank you. (Please see lines 151-152 on page 4.)

L129-139 - This section is confusing. There is no lead-in to discuss the connection of the research to your overarching theme within this section as you go straight to "The training program..." Please revise to introduce and draw better connections to how the studies are reflected within your theme of this section. 

Response: We have amended it accordingly. Thank you. (Please see lines 151-161 on page 4.)

L140-144 - Missing citations for these statements.

Response: We have amended it accordingly. Thank you.

Section 3.3 - Please indicate the number of studies you located in your review which focus on improving intellectual wellness due to team sports.

Response: We have amended it accordingly. Thank you. (Please see lines 276-277 on page 7.)

L235-247 - You state there are multiple studies that entail compelling evidence, however there no citations throughout. This section ends with only two references. Please revise.

Response: We have amended it accordingly. Thank you. (Please see lines 282-294 on page 7.)

Section 3.4 - Please indicate the number of studies you located in your review which focus on improving social wellness due to team sports.

Response: We have amended it accordingly. Thank you. (Please see lines 328-329 on page 8.)

L273-277 - Please revise these sentences.

Response: We have amended it accordingly. Thank you. (Please see lines 336-338 on page 8.)

L277-284 - The study including students within a Physical Education class needs more explanation as there are many different types of activities students participate within during physical education. For example, was this specifically a team sports class? Was the goals of this class specific to students developing social skills through team sport?

Response: We have amended it accordingly. Thank you. (Please see lines 341-353 on page 8.)

Section 3.5 - Please indicate the number of studies you located in your review which focus on improving spiritual wellness due to team sports.

Response: We have amended it accordingly. Thank you. (Please see lines 385-386 on page 9.)

L336-341 - Please cite.

Response: We have amended it accordingly. Thank you. (Please see lines 415 on page 10.)

Section 3.6 - Please indicate the number of studies you located in your review which focus on occupational wellness.

Response: We have amended it accordingly. Thank you. (Please see lines 427-428 on page 10.)

L358-363 - Please cite.

Response: We have amended it accordingly. Thank you. (Please see lines 446 on page 10.)

L352-357 - This study is focused on university students. Again, including more detail around the parameters of what population your review of interventions focused on would be beneficial.  

Response: We have amended it accordingly. Thank you. (Please see lines 433-439 on page 10.)

Overcoming Barriers Sections - I really enjoyed reading these sections, good overview for factors that limit sport participation.

Thank you.

Section 4.3 (L481-490) - Written awkward, please revise this section.

Response: We have amended it accordingly. Thank you. (Please see lines 573-580 on page 14.)

Submission Date

12 March 2024

Date of this review

12 Apr 2024 14:39:21

Round 2

Reviewer 1 Report

Comments and Suggestions for Authors

The article remains the same as it was originally written. None of the suggestions and recommendations of this reviewer have led to any text change, modification or improvement. It is not accepted for that reason. 

Comments on the Quality of English Language

Minor editing of English language required

Author Response

Thank you for your thorough review and the insightful comments provided regarding our manuscript titled "Team Sports as Catalysts for Holistic Student Wellness: Insights from a Narrative Review". We appreciate the time and effort you have devoted to evaluating our work.

Reviewer 3 Report

Comments and Suggestions for Authors

The authors have adequately addressed my comments for revisions.

Author Response

Thank you for your review and confirmation. We appreciate your guidance and support in improving our manuscript.

Round 3

Reviewer 1 Report

Comments and Suggestions for Authors

Dear authors

As I have already said in my last review there are no changes or improvementes in the document,  and for that reason, I can not certificate advances and palpable improvements in the manuscript. I understand their authors have their ideas and arguments, but their text need folow the reviewer recomendations, because if not, it is not possible materialize and confirm those improvements and new perspectives in the final document.  

Best regards

Comments on the Quality of English Language

Minor editing of English language required.

Author Response

1) The initial statement of the benefit of team sports for the whole world is very general. The justification does not specify any country of reference, nor is it even known if it is oriented to the Spanish educational system. As the study rightly points out, the differences between cultures are so significant that it looks general to approach it in terms of the world. It would be convenient to include in the justification a small diagnosis of the current situation of the school institutions regarding the analyzed sporting practice. Then, we would know why or what to consider or improve.

Response:

Team sports, encompassing activities where athletes collaborate to achieve shared objectives, are crucial in developing communication, conflict management, and problem-solving within a trusting team environment. However, the initial assertion that team sports universally benefit all educational systems is overly broad, lacking specificity regarding geographical or cultural contexts. A more nuanced approach is necessary given the significant cultural variations that influence sporting practices.

Thank you for your suggestions. We have revised the wording that was not appropriate. Please refer to lines 35-61.

2) What do the authors mean by the wheel of well-being? It is not an academic or scientific theory. Where does this theory come from? Mixing the emotional, the intellectual, the social and the spiritual seems like it could be more logical. They are very different dimensions. It is possible to understand the connection between the emotional, the intellectual and the social. Work could be oriented to these three spheres in a much more coherent and precise way. The occupational part of work is connected to the emotional. This part should be deleted to avoid turning the study into a catch-all for all dimensions. It is not necessary. On the other hand, even if we understand its relevance, the spiritual dimension does not directly connect to team sports and the promotion of mental and physical well-being. This connection to the spiritual appears forced. It is not necessary. This excess of relations and meanings calls the method of the wellness wheel into serious question. It is not necessary to quote this method. The relationship to mental, physical, emotional, intellectual, and social well-being is more than evident. One could name another more scientific and academic study, method or theory and delete this well-being wheel, which sounds good but tries to incorporate too many topics and needs more focus on the fundamentals.

Response:

  1. Thank you for your feedback regarding the Wellness Wheel. This model, first introduced by Sweeney and Witmer in 1991 and supported by the Global Wellness Institute, is well-established in health promotion. It provides a holistic framework for understanding the multifaceted nature of well-being, encompassing physical, emotional, intellectual, social, spiritual, and occupational dimensions. This study utilizes the Wellness Wheel to systematically examine the impact of team sports across these interconnected dimensions, demonstrating its relevance and applicability in enhancing student wellness.

Thank you for your suggestions. We have revised the wording that was not appropriate. Please refer to lines 47-51.

  1. Thank you for your thoughtful observations regarding the occupational and spiritual dimensions. Regarding occupational wellness, the research underscores that team sports cultivate vital skills such as teamwork and leadership, directly transferable to professional settings, enhancing occupational readiness and adaptability (Moustakas et al., 2023; McEwan et al., 2014 ). As for the spiritual dimension, team sports often extend beyond physical activity to promote a sense of community and shared purpose. This collective experience can significantly contribute to spiritual well-being by fostering a sense of belonging and personal fulfillment, which are crucial aspects of holistic health (Clarke, 2010; Roychowdhury, 2019). These elements underscore the multidimensional impact of team sports, validating their inclusion in the study.

Thank you for your suggestions. We have revised the wording that was not appropriate. Please refer to lines 79-86.

  1. Thank you for your critical insights concerning using the Wellness Wheel and its scope within our study. We acknowledge the concerns regarding the potential breadth of the model. However, employing the Wellness Wheel aims to underscore the holistic benefits of team sports, highlighting how these activities impact various aspects of a student's life. To strengthen our analysis and maintain academic rigor, We will further clarify the interconnectivity between the dimensions—showing how physical, emotional, social, intellectual, spiritual, and occupational wellness are interlinked and collectively contribute to comprehensive student development.

Thank you for your suggestions. We have revised the wording that was not appropriate. Please refer to lines 439-456.

3) The study contains some allusions to the sociology of sport and the different sporting practices according to different cultures. How can minority sports such as rugby, which appears in the study, be incorporated into school environments? Football is still predominant in many countries for cultural reasons, and recently, there has been a boom in gymnastics. This fact, so actual, is not mentioned in the publication. What team sports are in fashion, or does it depend on the country, and how do young people view team sports? If they are so positive, why don't they generate motivation? This question does not appear in the conclusions either. What can we do? What are the future lines of research to improve the incorporation of team sports in educational institutions? Why should this situation change if it has changed little in the last decades? If it is changing, in what sense? These answers do not appear in the conclusion. The conclusion that team sports are beneficial, which appears, is undeniable.

Response:

Thank you for your suggestions. We have attempted to rewrite the conclusion section in hopes that the manuscript will allow readers to understand the points of view expressed in this paper quickly. Please refer to lines 600-623.